# Fibroblast Growth Factor 23: Potential Marker of Invisible Heart Damage in Diabetic Population

**DOI:** 10.3390/biomedicines11061523

**Published:** 2023-05-25

**Authors:** Anna Kurpas, Karolina Supel, Paulina Wieczorkiewicz, Joanna Bodalska Duleba, Marzenna Zielinska

**Affiliations:** 1Department of Interventional Cardiology, Medical University of Lodz, 251 Pomorska Street, 92-213 Lodz, Poland; a.lipinska.a@gmail.com (A.K.); paulina.wieczorkiewicz@umed.lodz.pl (P.W.); marzenna.zielinska@umed.lodz.pl (M.Z.); 2Diabetes Outpatient Clinic ‘Poradnia Nowa’, 90-631 Lodz, Poland; j.bodalska.duleba@gmail.com

**Keywords:** fibroblast growth factor 23, two-dimensional speckle-tracking echocardiography, left ventricular diastolic dysfunction, diabetes mellitus

## Abstract

Two-dimensional speckle-tracking echocardiography (2DSTE) detects myocardial dysfunction despite a preserved left ventricular ejection fraction. Fibroblast growth factor 23 (FGF23) has become a promising biomarker of cardiovascular risk. This study aimed to determine whether FGF23 may be used as a marker of myocardial damage among patients with diabetes mellitus type 2 (T2DM) and no previous history of myocardial infarction. The study enrolled 71 patients with a median age of 70 years. Laboratory data were analyzed retrospectively. Serum FGF23 levels were determined using a sandwich enzyme-linked immunosorbent assay. All patients underwent conventional echocardiography and 2DSTE. Baseline characteristics indicated that the median time elapsed since diagnosis with T2DM was 19 years. All subjects were divided into two groups according to left ventricular diastolic function. Individuals with confirmed left ventricular diastolic dysfunction had significantly lower levels of estimated glomerular filtration rate and higher values of hemoglobin A1c. Global circumferential strain (GCS) was reduced in the majority of patients. Only an epicardial GCS correlated significantly with the FGF23 concentration in all patients. The study indicates that a cardiac strain is a reliable tool for a subtle myocardial damage assessment. It is possible that FGF23 may become an early diagnostic marker of myocardial damage in patients with T2DM.

## 1. Introduction

Diabetes mellitus (DM) poses a serious challenge to the world’s health. Despite global efforts to tackle the diabetes pandemic, the number of adults living with DM is on the rise. The International Diabetes Federation reported that the number of adults with DM has already surpassed 537 million and is predicted to rise to 643 million by 2030 [1]. Even more frightening, is the fact that there are another 541 million people with impaired glucose tolerance, who are at high risk of type 2 diabetes mellitus (T2DM). T2DM accounts for approximately 90% of all cases of DM [2].

Cardiovascular disease (CVD) is the most prevalent cause of morbidity and mortality in a diabetic population [3]. DM is associated with a 2- to 3-fold increased risk of myocardial infarction (MI) and stroke [4]. The dreadful combination of DM and ischemic heart disease (IHD) has always put a severe strain on diabetologists and cardiologists. An increased prevalence of IHD in patients with DM contributes to the greater incidence of heart failure (HF). The risk of HF in diabetic patients is more than twice that of the nondiabetic population [5,6].

Nearly 50 years ago, Rubler et al. introduced the concept of diabetic cardiomyopathy (DCM), which is considered a primary myocardial disease in diabetic patients [7,8]. DCM was defined as abnormal cardiac structure and performance in the absence of IHD, arterial hypertension (HA), or significant valvular disease. Despite the importance of this entity, the mechanisms underlying DCM pathogenesis have remained poorly elucidated. Dysregulated glucose and lipid metabolism causes increased oxidative and inflammation stress, which both mediate pathological cardiac remodeling, characterized by left ventricular (LV) concentric hypertrophy and increased myocardial fibrosis [9]. Subsequently, the gradual decline in LV function (diastolic often precedes systolic) may be observed [10]. Therefore, a diabetic population is particularly susceptible to a long-standing subclinical myocardial dysfunction before development of overt HF.

DM claims lives and triggers disability. Patients with DM have unfavorable prognoses due to multiple diabetic complications and frequent comorbidities, worse cardiovascular (CV) outcomes, and higher rates of hospitalization compared to their healthy counterparts [5,6,11]. Numerous studies indicate that poor glycemic control is associated with an increased risk of exacerbation of IHD and HF [12,13]. DM combined with cardiac diseases is frequently related to a poorer quality of life and the burden of high medical costs. 

Transthoracic echocardiography (TTE) is the most fundamental method for a cardiac function evaluation of patients with DM. Although reduced left ventricular ejection fraction (LVEF) indicates HF among symptomatic patients, a considerable proportion of patients with HF still have preserved LVEF [14]. The latter cannot be efficiently detected by traditional echocardiographic parameters. Hence, development of other techniques, particularly noninvasive, objectively identifying myocardial dysfunction is highly desirable.

In 1973, Mirsky and Parmley described the concept of evaluating myocardial stiffness by using a measure of deformation (i.e., strain) [15]. Myocardial strain was defined as the percent change in the length of a myocardial segment relative to its resting length and considered an indicator of LV function [16]. Two-dimensional speckle-tracking echocardiography (2DSTE) is a new technique, which enables reliable evaluation of regional deformation in three directions: longitudinal, circumferential, and radial [17]. Due to 2DSTE, it is feasible to identify patients with a preserved LVEF, no previous history of CVD, and asymptomatic LV systolic or diastolic dysfunction.

In recent years, fibroblast growth factor 23 (FGF23) has gained wide attention in many fields of medicine and has become a promising biomarker linking chronic kidney disease (CKD) with CV morbidity and mortality [18,19]. FGF23, as an endocrine-acting phosphaturic hormone, plays a pivotal role in calcium-phosphate metabolism and interorgan signaling. Apart from its physiologic actions, an elevated FGF23 level is also associated with pathologic effects, such as left ventricular hypertrophy, and mediates cardiac remodeling [20]. Some data indicate that FGF23 triggers production of inflammatory markers (e.g., transforming growth factor β), which promotes development of myocardial fibrosis [21,22]. Multiple publications have indicated that FGF23 is a novel biomarker of CV risk; however, most of them have promoted a clear relationship between FGF23 level and CV mortality and morbidity in patients with HF rather than IHD [23,24]. In our previous study, we reported consistent results with recent findings. We did not provide the evidence for any correlation between FGF level and either overt IHD (33% of study population) or LVEF (56%, interquartile range (IQR) 52–60) [25]. Therefore, in furtherance of our earlier research, we aimed to determine whether FGF23 may be used as a marker of early myocardial damage among patients with long-standing T2DM and no previous history of MI. The utility of FGF23 was assessed with reference to the global longitudinal (GLS) and global circumferential strain (GCS). An early detection of DCM is pivotal to the enhancement of outcomes in a diabetic population.

## 2. Materials and Methods

### 2.1. Study Design and Population

For the present analysis, we included 71 consecutive patients. Data of all individuals were obtained from a Diabetes Outpatient Clinic (Lodz, Poland) database between 2019 and 2021. Patients had the following inclusion criteria: (1) T2DM duration of >10 years; (2) regular follow-up care by a mutual diabetologist; and (3) hemoglobin A1c (HbA1c) ≤ 8% were enrolled in the study, whereas subjects with any of the following conditions were excluded: (1) age < 18 years; (2) previous history of MI; (3) active infection; (4) malignant tumor; (5) no electronic TTE data available for retrospective analysis; or (6) missing key clinical data. Afterwards, participants were divided into two groups according to a diagnosis of left ventricular diastolic dysfunction (LVDD): Group 1 (non-LVDD, n = 46) or Group 2 (LVDD, n = 25). A flowchart of the study is shown in Figure 1. 

Demographic and clinical characteristics were collected during a structured interview, and included, among others: sex, age, height, body weight, body mass index (BMI), T2DM duration, and smoking history. Data related to comorbidities (e.g., HA, arrythmias), cardiovascular hospitalization, and family history of heart disease or DM were obtained from all participants as well. Diaries of self-control were used to verify home blood pressure monitoring. Laboratory examination data, such as HbA1c (%), creatinine (Cr; mg/dL), estimated glomerular filtration rate (eGFR) (mL/min/1.73m^2^), total cholesterol (TC) (mg/dL), high-density lipoprotein cholesterol (HDL-C) (mg/dL), low-density lipoprotein cholesterol (LDL-C) (mg/dL), non-high-density lipoprotein cholesterol (non-HDL-C) (mg/dL), and triglycerides (TG) (mg/dL), were analyzed retrospectively. 

The study protocol was in accordance with the Declaration of Helsinki and approved by the Ethics Committee of the District Medical Chamber in Lodz (No. K.B.-10/18, 11 April 2018). Written informed consent was obtained for inclusion of the patients.

### 2.2. FGF23 Measurement

Serum levels of intact FGF23 (iFGF23) were measured using a sandwich enzyme-linked immunosorbent assay (ELISA) with a FGF23 ELISA kit (SunRed Biological Technology, Shanghai, China). All blood samples were collected in the morning after an overnight fast (minimum 10 h), during previously scheduled appointments at the Diabetes Outpatient Clinic, and stored at −80 °C before analysis. The reference range for serum iFGF23 level in healthy adults is 8.2–54.3 pg/mL [26].

### 2.3. Conventional Echocardiography

All patients underwent two-dimensional (2D) TTE using a commercially available ultrasound system (Vivid E9, General-Electric Healthcare, Horten, Norway) equipped with a 3.5-Mhz transducer. Participants were evaluated with 2D, M-mode, color Doppler, pulse Doppler, and continue Doppler echocardiographic examinations in the left lateral decubitus position. To achieve optimal image quality, we adjusted gain, compression, depth, and sector width. The images were acquired in the apical (four- and two-chamber and apical long-axis views) and parasternal views (long- and short-axis views) at the basal, papillary, and apical LV levels at high frame rates of 70–90 frames/s.

LV volumes were measured from the apical four- and two-chamber views. LVEF was calculated in the apical four-chamber view using the modified Simpson’s rule. LVDD was defined in accordance with the most recent guidelines [27]. To determine whether a diastolic function is normal or abnormal, analysis of four variables with the following cutoff values was performed: (1) annular e′ velocity (septal e′ < 7cm/s, lateral e′ < 10cm/s); (2) average E/e′ ratio > 14; (3) peak tricuspid regurgitation velocity > 2.8 m/s; and (4) left atrial volume index > 34 mL/m^2^. If more than two parameters met the above-mentioned cutoff values, LVDD was diagnosed, as shown in Figure 2.

### 2.4. Two-Dimensional Speckle-Tracking Echocardiography

LV function was further evaluated with 2DSTE using a software package (EchoPAC, Horten, Norway, version 201 software, General-Electric Medical Systems) by two experienced and independent observers. In each plane, three consecutive cardiac cycles were captured during a breath hold at the end-expiration and stored in a cine-loop format for offline analysis. 2DSTE is a relatively new technique of cardiac imaging based on frame-by-frame tracking of ultrasonic speckles in gray scale. It allows accurate assessment of myocardial deformation in two dimensions. Two different patterns [i.e., circumferential strain (CS) and longitudinal strain (LS)] may be evaluated using the apical and parasternal short-axis views. LS shows systolic shortening in the long-axis plane, while CS represents systolic shortening but in a short-axis plane. During systole, LS and CS are anticipated to be negative values. Moreover, we analyzed CS in three myocardial layers: epicardium, mid-wall, and endocardium, under the assumption that the alterations among diabetic population may appear layer-specific. The software automatically divided the region of interest (ROI) into six equal segments and provided the time–strain curves for each myocardial segment. If necessary, ROI was manually adjusted using a point-and-click approach to enhance tracking quality. Curves of regional and global peak systolic strain, previously calculated by segmental averaging, were obtained. The reference range for GLS and GCS strain values are based on Nagata et al. [28].

### 2.5. Statistical Analysis

All statistical analyses were performed using STATISTICA v. 14 software (StatSoft Polska, Kraków, Poland). A *p*-value of 0.05 was used as the threshold of statistical significance. The normality assumption was verified using the Shapiro–Wilk test. Categorical variables are expressed as the number of observations (N) with the corresponding percentages (%), whereas quantitative variables as median and IQR. Pearson’s χ^2^ test was used to determine differences between categorical variables. If the number of cases were less than 5, Yates’s correction for continuity was used. Continuous variables were analyzed with a nonparametric test. The Mann–Whitney U test was used to compare two independent trials. Spearman’s rank correlation coefficient was used for assessment of correlation strength.

## 3. Results

A total of 71 individuals with T2DM and no prior history of MI were enrolled in the study. There were 36 women (51%) and 35 men (49%) with a median age of 70 years (IQR 66–74). Baseline characteristics indicated that the median time elapsed since diagnosis with T2DM was 19 years (IQR 13–24). According to the results, the median BMI of the study population was 29.7 kg/m^2^. Patients were diagnosed with various diseases, among others: arterial hypertension (83%), atrial fibrillation (AF) (5%), and stroke (7%). Furthermore, we assessed the frequency of chronic complications of DM, such as retinopathy (17%), neuropathy (10%), and diabetic foot syndrome (4%). Nearly two-thirds (66%) of the patients were either current or former smokers. As many as 15% of the patients were hospitalized due to CVD at least once in their lifetime. Family history of heart disease and DM were, respectively, reported in 49% and 66% patients. Table 1 presents demographic and clinical data according to the presence or absence of LVDD. As shown, there are no statistically significant differences between the two groups of patients.

Table 2 demonstrates the laboratory test results according to an LV diastolic function status. In our study, we determined the concentration of FGF23, HbA1c, renal parameters, and a lipid profile.

FGF23 was elevated in all patients (256 pg/mL, IQR 214–567). Interestingly, there was no significant difference in its levels between patients with LVDD and normal LV diastolic function (255 pg/mL, IQR 206–567; 268 pg/mL, IQR 232–380; *p* = 0.918), as shown in Figure 3. Individuals with confirmed LVDD appeared to have significantly higher concentrations of HbA1c and lower levels of eGFR (7%, IQR 6.7–7.7; 80 mL/min/1.73 m^2^, IQR 64–101) compared to subjects with normal diastolic function (6.5%, IQR 6.3–7.2; 101 mL/min/1.73 m^2^, IQR 76–102). Furthermore, none of the lipid parameters correlated with either normal or abnormal LV diastolic function. A lipid profile included measurements of TC (155 mg/dL, IQR 129–187), HDL-C (51 mg/dL, IQR 43–62), LDL-C (73 mg/dL, IQR 54–98), non-HDL-C (102 mg/dL, IQR 77–126), and TG (121 mg/dL, IQR 90–169).

All patients underwent TTE evaluation. The echocardiographic findings are summarized in Table 3. Individuals with LVDD had higher values of LVMI (109 g/m^2^, IQR 100–121; *p* = 0.003), IVSs (17 mm, IQR 16–19; *p* = 0.014), and IVSd (13mm, IQR 11–13; *p* = 0.002). However, no correlation was found between either normal or abnormal LV diastolic function and the following variables: LVEF, LVESV (left ventricular end systolic volume), LVEDV (left ventricular end diastolic volume), left atrial volume, LAVI (left atrial volume index), TAPSE (tricuspid annular plane systolic excursion), and RVOT (right ventricular outflow tract) proximal diameter.

The parameters derived from 2DSTE are listed in Table 4. After considering age dependency, average GCS was diminished in the majority of patients. Average GCS was −16.4%, endocardial GCS was −24.8%, whereas epicardial GCS was considerably reduced and amounted to −9.2%. According to Nagata et al., the reference ranges for average, endocardial, and epicardial GCS are, respectively, as follows: −21.2% ± 2.1, −29.3% ± 2.9, and −15.6% ± 1.9 [28]. GLS was within the normal range, which is −19.4% ± 1.7 [28]. Contrary to our expectations, there were no significant differences in GLS and GCS between patients with LVDD and normal LV diastolic function. Even the analysis of a layer-specific GCS did not reveal any correlation with LV diastolic function. The examples of LV CS curves of endocardium, mid-wall, and epicardium from papillary muscles level are illustrated in Figure 4.

This study also assessed the utility of FGF23 with reference to cardiac strain. It revealed that an epicardial GCS correlated with FGF23 level regardless of LV diastolic function status (*p* = 0.041), as shown in Figure 5. Such correlation was not found in the following subgroups, patients with LVDD and with normal diastolic function. None of the other strains correlated with the FGF23 levels either in the study population or in patients with LVDD or with normal diastolic function.

Distribution of GLS, endocardial and epicardial GCS values according to a LV diastolic function status are demonstrated in Figure 6A–C.

## 4. Discussion

The overwhelming majority of publications indicate a significant correlation between an elevated level of FGF23 and increased CV risk in CKD. However, little is known about the relationship between FGF23 and CVDs in a diabetic population, particularly without kidney failure. Furthermore, most researchers have posited a clear association between FGF23 concentration and CV risk in the subjects with HF rather than IHD. Our study included patients with long-standing T2DM, relatively good glycemic control, preserved renal function, and no previous history of myocardial infarction. Due to a careful selection of the target group, the interference of confounding factors was reduced to a certain extent. The main goal of the research was to determine whether FGF23 may be used as an early marker of myocardial damage among patients with T2DM and no previous history of myocardial infarction. To the authors’ best knowledge, this is the first study to investigate the correlation between FGF23 and a cardiac strain in a diabetic population. The study indicates that a cardiac strain is a reliable tool for a subtle myocardial damage assessment. We have proven that myocardial injury in patients with long-standing T2DM is layer-specific and starts from the epicardium. Furthermore, we performed an in-depth analysis showing that an elevated level of FGF23 is significantly associated with a reduced value of epicardial GCS among our patients. It is possible that FGF23 may become an early diagnostic marker of myocardial damage in patients with long-standing T2DM.

T2DM is a metabolic disease associated with a considerably higher risk of HF and CV mortality, even in the absence of IHD. Patients with T2DM and no previous history of IHD have an equal CV risk compared to a nondiabetic population with a prior MI. Pathophysiological links between T2DM and CVD have not been fully elucidated. It is known that DM has a profound effect on a CV system, causing extracellular matrix remodeling, increased oxidative stress, and endothelial cell dysfunction [29,30,31,32]. Over time, LV remodeling, myocardial fibrosis, and histological alterations lead to LVDD, which usually precedes systolic dysfunction. Eventually, one observes the development of overt HF. Concentric LV remodeling has been considered an adverse prognostic marker of CV events [33,34]. Some data also indicate that LV remodeling results from renin-angiotensin-aldosterone system activation, microangiopathy, inflammatory cytokines, or CV autonomic neuropathy [35,36]. Early myocardial injury among patients with T2DM is typically slow, insidious, and symptoms are not specific [37,38]. There is an increased risk of hospitalization and adverse outcomes.

Heart failure with preserved ejection fraction (HFpEF) is an urgent health and social problem. It is estimated that HFpEF accounts for nearly 50% of all HF cases [14]. It is possible that the high prevalence of HFpEF results from a greater awareness of physicians, but especially the ageing population and widely known risk factors, such as advanced age, HA, obesity, or metabolic syndrome. Its complex pathophysiology has not been fully elucidated; therefore, we may find multiple definitions in the literature. According to the earlier guidelines, LVDD was required to confirm HFpEF and responsible for HF manifestation. On the one hand, the CHARM echocardiographic substudy showed that LVDD was observed in 67% of patients, whereas only moderate and severe LVDD (44%) was a significant and independent predictor of adverse outcome [39]. On the other hand, a community-based study in Olmsted County (Minnesota, USA) demonstrated that among 2042 randomly selected residents, LVDD was confirmed in 28%, but only 2.2% were diagnosed with HF [40]. Diagnosis of HFpEF might be challenging, even for HF specialists. However, identification of patients, among LVDD cases, who will subsequently develop HF seems particularly difficult.

LVEF assessed in TTE is a cornerstone of cardiac function evaluation. Nonetheless, we must bear in mind that LVEF may remain preserved in the early stages of many heart diseases, despite actual impaired myocardial contractility. Many scientists suggest that the prognosis of patients with reduced and preserved LVEF is similar [14,41]. Myocardial strain is based on the speckle-tracking method and has been acknowledged as a more accurate tool for myocardial function assessment than conventional echocardiography. Park et al. conducted a study of 4172 patients with acute HF [42]. Approximately 40% of them died within 5 years. They demonstrated that GLS was a better prognostic marker than LVEF. No difference was observed in LVEF between still living individuals and those who had died, whereas GLS was reduced in deceased patients. The same researchers carried out a study, which included 355 patients with LVEF of ≥50% and without significant LVDD, to evaluate GLS value as a HF predictor [43]. Impaired GLS was defined as <16%. They showed that 28% of patients with acute HF and preserved LVEF as well as elevated levels of natriuretic peptides had no LVDD. The utility of conventional echocardiography appeared limited in this group of patients, because according to the contemporary definition of HFpEF, they would not have been classified as having HF [44]. The analysis of GLS during resting TTE indicated that the individuals with an impaired GLS had a worse prognosis. Data concerning GCS as a prognostic marker are rather poor. Collaborators from South Korea and Australia published an article implying that GCS is a better predictor of adverse cardiac events (readmission for HF or cardiac death) than LVEF and GLS [45]. They evaluated data of 201 subjects hospitalized for acute HF. Pezel et al. conducted a study of 1506 patients, of whom 122 were diagnosed with severe IHD and 91 had confirmed HF during follow-up (15.9 years [12.9–16.6]) [46]. They were the first to assess the prognostic value of layer-specific regional CS in the general population. Eventually, they proved that a layer-specific regional CS may be an independent predictor of incidental HF and severe IHD. Skaarup and colleagues demonstrated that endocardial GCS, rather than epicardial GCS, is independently related to incidental HF in the general population and may be used as a marker of early cardiac pathology [47]. Our study likewise indicates that a cardiac strain is a better tool for a subtle myocardial assessment than LVEF. In the literature, one may find rather inconsistent data regarding the best evaluated strain parameter. This issue clearly requires further research.

Interestingly, there are a few articles presenting the relationship between myocardial strain and DM; however, the great majority of them concern LV LS, while CS was not explored or did not show any difference in a diabetic population. Flores-Ramírez et al. conducted a study of 121 diabetic patients including: 14 individuals with mildly reduced LVEF, 76 diabetics with preserved LVEF, and 31 controls [48]. They showed that the diabetics had lower GLS than the controls; however, no difference was found in GLS in the diabetics with preserved LVEF compared to the controls. Chinese researchers analyzed the data of 247 patients with T2DM and no prior history of CV complications [49]. They concluded that impaired GLS in this population was an independent predictor of CV events, such as acute coronary syndrome, cerebrovascular stroke, CV death, and hospitalization for HF. A recent concept has been based on a multilayer analysis of myocardial deformation, which relates to an anatomical aspect. Tadic et al. evaluated 146 individuals (44 controls, 48 patients with DM, and 54 patients with DM and HA) who underwent 2DSTE [50]. The research led to the following conclusions: (1) GLS and GCS gradually worsened from controls, through patients with DM, to those with both DM and HA; (2) all layers of myocardium were affected by DM and HA; (3) progressive deterioration of impaired GLS and GCS in mid-wall and epicardial layers were observed; and (4) HA exerted an additional deleterious effect on LV deformation in patients with DM. Our findings unequivocally suggest that an epicardium is the first affected layer of myocardium in patients with T2DM. Apparently, large clinical trials are highly desired to dispel any doubts related to GLS or GCS as potential predictors of adverse outcomes in both the general and diabetic populations. It is possible that the multilayer analysis of myocardial deformation may prove beneficial for CV risk stratification as well.

Several publications indicate that the underlying pathophysiology of systolic and diastolic dysfunction is aberrant calcium homeostasis [51,52]. Adeniran et al. developed a biophysically detailed model of HFpEF to explore calcium mishandling in LVDD [53]. They demonstrated that HFpEF triggers systolic calcium level decline, which subsequently leads to reduction in contractile reserve. Simultaneously, they noted an increase in diastolic calcium concentration, which results in prolonged relaxation and increased filling pressure. Interestingly, mounting evidence pointed to FGF23 as a regulator of intracellular calcium and cardiac contractility. American scientists showed that an abrupt rise in FGF23 level affects primary cardiomyocytes causing a significant increase in intracellular calcium [54]. Furthermore, prolonged exposure to FGF23 triggered calcium overload. They suggested that calcium may appear to be a key link between increased FGF23 level, long-term cardiac remodeling, hypertrophy, and finally HF. Numerous publications demonstrated that elevations in a circulating FGF23 have a clear relationship with HF. For example, Robinson-Cohen et al. included 6413 participants without CVD, and found a significant correlation between FGF23 level and HF risk [23]. Another large study showed that an increased level of FGF23 was associated with subclinical cardiac disease, new HF, and a 14% greater risk of IHD [24]. Data concerning FGF23 level in a diabetic population are rather poor and inconsistent, especially in terms of patients with preserved renal function [55,56]. In the literature, there are a few hypotheses explaining the relationship between elevated FGF23 level and diabetes mellitus [57,58,59,60,61,62,63,64].

One of the questions that interested us concerned a relationship between FGF23 and myocardial strain in a diabetic population. Our results revealed that epicardial GCS correlated significantly with elevated FGF23 level in a T2DM population, with a preserved LVEF, but confirmed LVDD. It is noteworthy that patients enrolled in our study had relatively good glycemic control, despite long-standing T2DM, preserved kidney function, and no previous history of MI. To our knowledge, no studies have so far explored this issue. Patel et al. evaluated the relationships of baseline serum FGF23 with cardiac magnetic resonance (CMR) measurements at a 10-year follow-up [65]. They demonstrated that in a multi-ethnic, community-based cohort of 2276 participants, baseline FGF23 concentrations were independently associated with higher LV mass, reduced GCS, mid-wall CS, and lower left atrium function; GLS was not assessed. Belgian researchers prospectively analyzed the data of 143 patients with HFpEF and 31 controls of similar age and gender [66]. They showed that: (1) FGF23 level was significantly elevated in HFpEF patients compared to controls; (2) FGF23 correlated with fibrosis estimated by extracellular volume measured by CMR T1 mapping; and (3) higher FGF23 concentration was associated with some proinflammatory co-morbidities, such as CKD, DM, and AF.

According to the current state of the art findings and our findings, the incorporation of a new technique, such as 2DSTE in combination with FGF23, would increase the diagnostic sensitivity of subclinical myocardial dysfunction in asymptomatic patients with T2DM. The question arises whether GLS or GCS has better predictive value of adverse CV outcome in this population. Undoubtedly, GLS has become the most widely used strain parameter and has been considered a better risk predictor than LVEF. In the literature, one may find rather poor information about GCS as a tool for CV risk stratification, particularly in a diabetic population. FGF23 is another potent predictor of CV risk. Our results revealed that GLS and GCS were both diminished in all patients. However, only epicardial GCS correlated significantly with an elevated level of FGF23 in T2DM patients. Data from clinical trials are highly awaited in order to evaluate the role of GCS and FGF23 in patients with T2DM.

## 5. Study Limitations

The present investigation has certain limitations. Our study was limited by its sample size. Furthermore, FGF23 level might have been affected by numerous conditions. Moreover, its measurement might have been affected by technical problems. Finally, the normal range of FGF23 and GCS has not been established yet; therefore, their practical usefulness is limited. 

## 6. Conclusions

Diabetes mellitus is a global epidemic, frequently associated with an increased CV risk, resulting in significant cardiac morbidity and mortality. Early detection of HF is of paramount importance to a preventive approach aimed at enhancing outcomes in this population. Our study indicates that a cardiac strain is a reliable tool for a subtle myocardial damage assessment. Interestingly, the myocardial injury in patients with T2DM was found to be layer-specific and started from the epicardium. Furthermore, we showed that an elevated level of FGF23 was significantly associated with a reduced value of epicardial GCS among patients with long-standing T2DM. It is possible that FGF23 may become an early diagnostic marker of myocardial damage in patients with long-standing T2DM. Further larger investigations are essential to validate our findings. Estimation of calcium concentration would be a worthwhile subject of future research. 

## Figures and Tables

**Figure 1 biomedicines-11-01523-f001:**
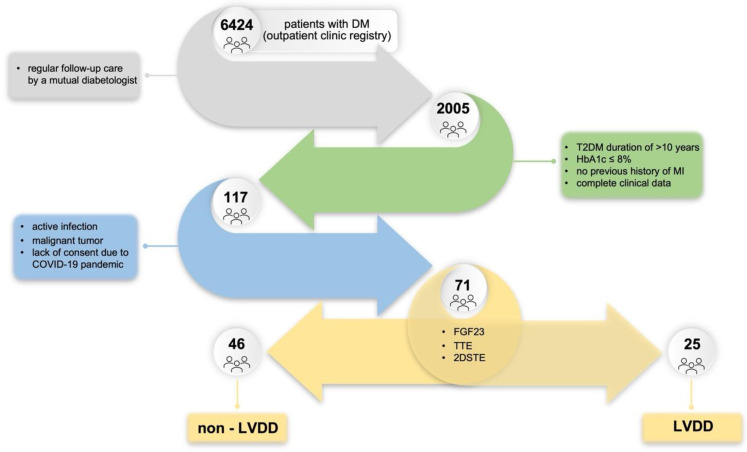
Study flow chart. Abbreviations: DM, diabetes mellitus; T2DM, type 2 diabetes mellitus; HbA1c, hemoglobin A1c; COVID-19, coronavirus disease 2019; FGF23, fibroblast growth factor 23; TTE, transthoracic echocardiography; 2DSTE, two-dimensional speckle-tracking echocardiography; LVDD, left ventricular diastolic dysfunction.

**Figure 2 biomedicines-11-01523-f002:**
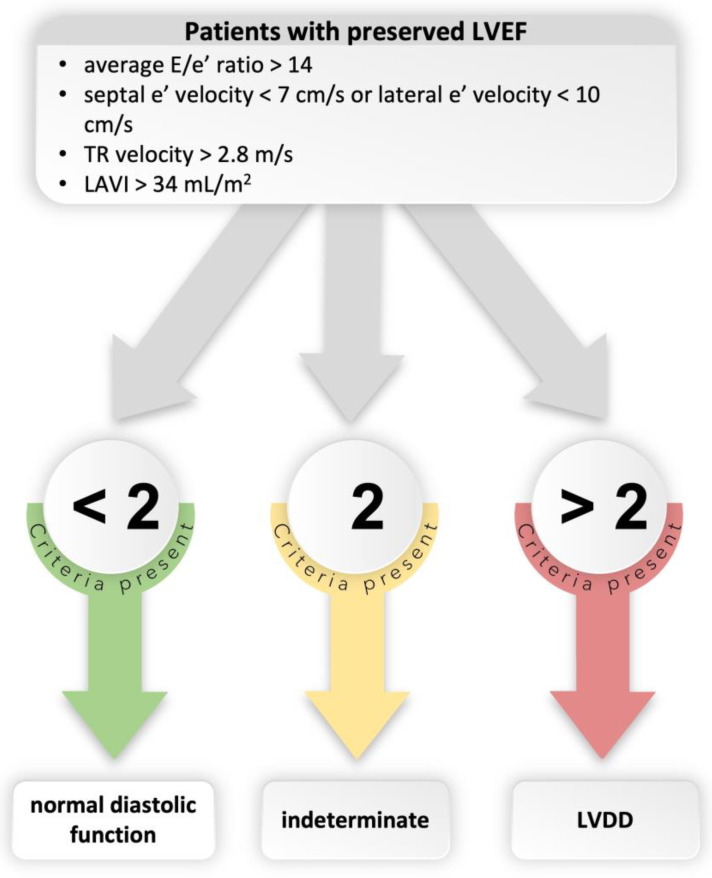
Algorithm for diagnosis of LVDD in patients with preserved LVEF. Abbreviations: LVEF, left ventricular ejection fraction; TR, tricuspid regurgitation; LAVI, left atrial volume index; LVDD, left ventricular diastolic dysfunction.

**Figure 3 biomedicines-11-01523-f003:**
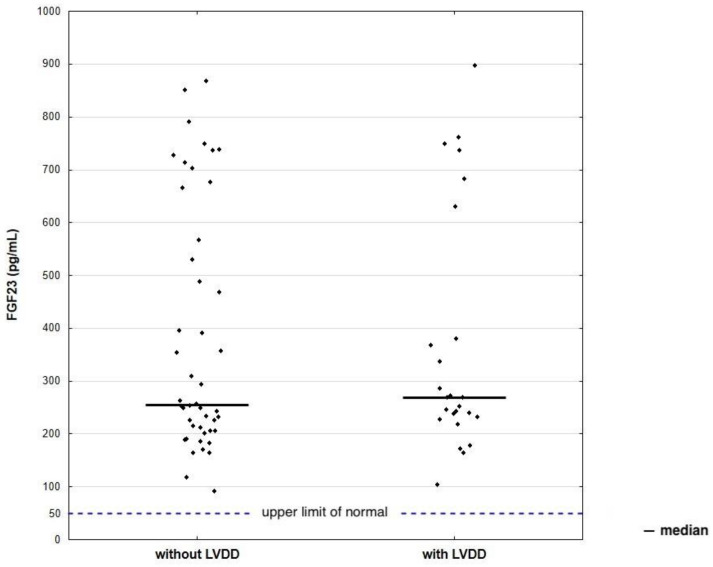
Scatter plot of FGF23 levels according to LV diastolic function status. Abbreviations: LVDD, left ventricular diastolic dysfunction; FGF23, fibroblast growth factor; LV, left ventricular.

**Figure 4 biomedicines-11-01523-f004:**
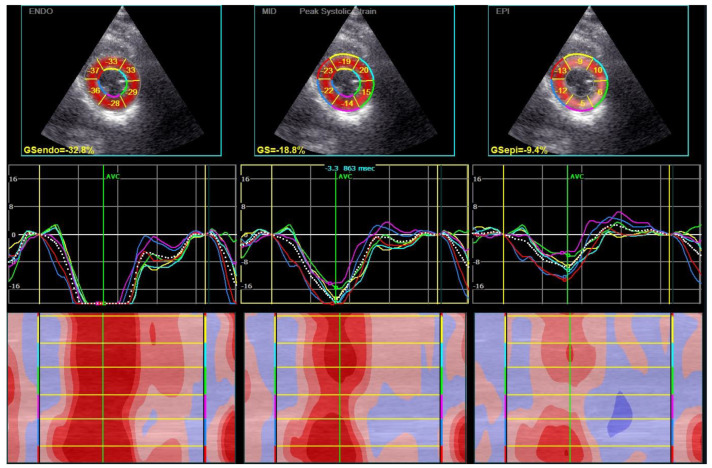
Layer–specific (endocardial, transmural, epicardial layer) circumferential global and segmental strain curves obtained from 2DSTE from papillary muscles level. Abbreviations: 2DSTE, two–dimensional speckle–tracking echocardiography.

**Figure 5 biomedicines-11-01523-f005:**
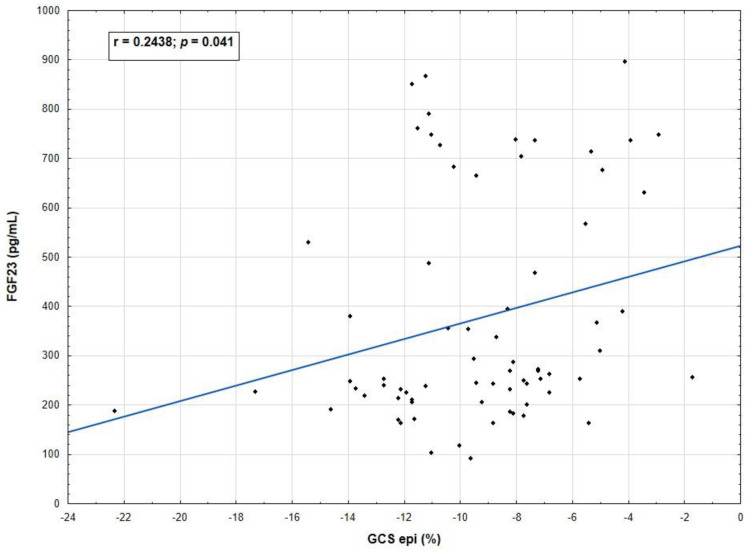
Spearman correlation between epicardial GCS values and FGF23 levels in patients with long–standing type 2 diabetes mellitus. Abbreviations: GCS, global circumferential strain; FGF23, fibroblast growth factor.

**Figure 6 biomedicines-11-01523-f006:**
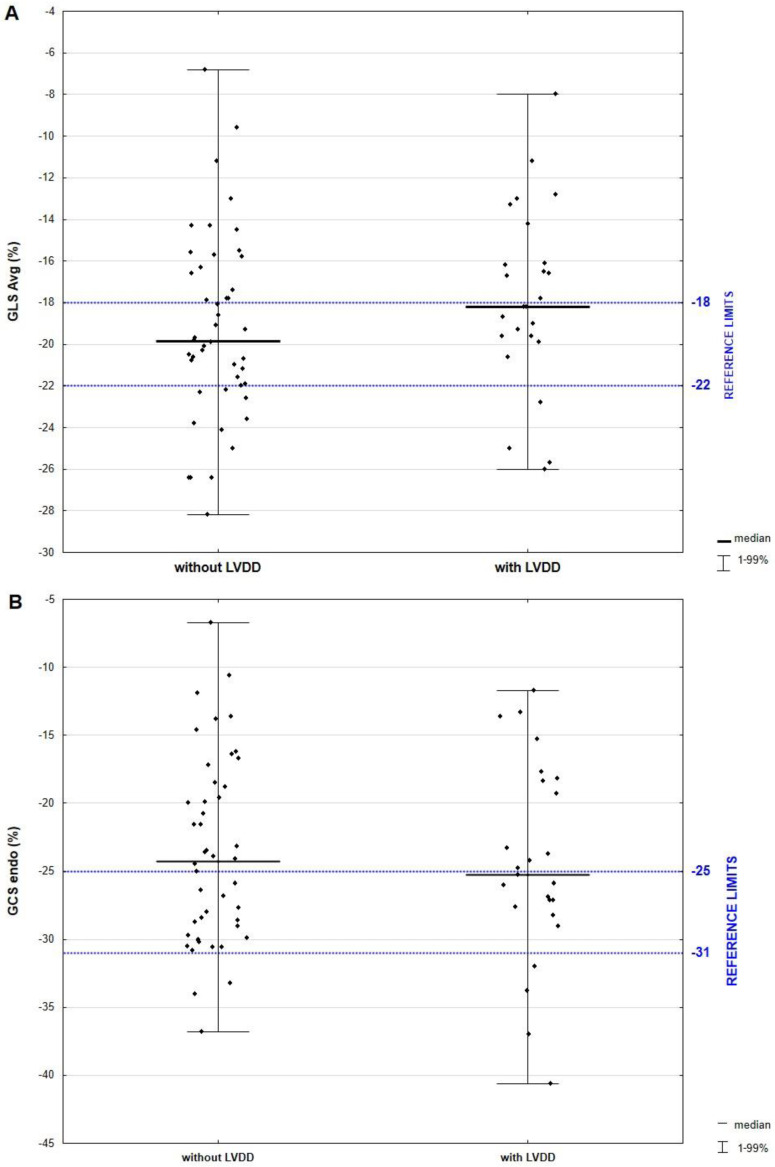
Scatter plot of average GLS (**A**), GCS endocardial (**B**), and GCS epicardial (**C**) levels according to LV diastolic function status. Abbreviations: LVDD, left ventricular diastolic dysfunction; GLS, global longitudinal strain; GCS, global circumferential.

**Table 1 biomedicines-11-01523-t001:** Baseline characteristics.

Variable	Total N = 71	Without LVDD N = 46	With LVDD N = 25	*p* Value
Age, years	70 (66–74)	67.5 (65–74)	70 (69–74)	0.061
Female sex	36 (51)	22 (49)	14 (56)	0.511
T2DM duration, years	19 (13–24)	18 (13–22)	19 (14–27)	0.446
BMI, kg/m^2^	29.7 (25–33)	29 (25–33)	29.7 (27–33)	0.572
Arterial hypertension	59 (83)	37 (80)	22 (88)	0.631
Stroke	5 (7)	2 (4)	3 (12)	0.472
Atrial fibrillation	4 (5)	3 (7)	1 (4)	0.921
Diabetic retinopathy	12 (17)	8 (17)	4 (16)	0.856
Diabetic neuropathy	7 (10)	4 (9)	3 (12)	0.977
Diabetic foot syndrome	3 (4)	1 (2)	2 (8)	0.583
Cardiovascular hospitalization	11 (15)	8 (17)	3 (12)	0.798
Family history of heart disease	35 (49)	24 (52)	11 (44)	0.511
Family history of diabetes mellitus	47 (66)	29 (63)	18 (72)	0.446
Current and former smokers	47 (66)	30 (65)	17 (68)	0.979
Nonsmokers	24 (34)	16 (35)	8 (32)	0.979

Note: Data are expressed as median (interquartile range) or number (%). Abbreviations: LVDD, left ventricular diastolic dysfunction; T2DM, type 2 diabetes mellitus; BMI, body mass index.

**Table 2 biomedicines-11-01523-t002:** Laboratory test results.

Variable	Total N = 71	Without LVDD N = 46	With LVDD N = 25	*p* Value
FGF23, pg/mL	256 (214–567)	255 (206–567)	268 (232–380)	0.918
Cr, mg/dL	0.83 (0.73–0.96)	0.79 (0.7–0.92)	0.91(0.79–1.04)	0.022
eGFR, mL/min/1.73 m^2^	88 (75–102)	101 (76–102)	80 (64–101)	0.029
HbA1c, %	6.8 (6.4–7.4)	6.5 (6.3–7.2)	7 (6.7–7.7)	0.045
TC, mg/dL	155 (129–187)	157 (132–194)	140 (125–176)	0.327
HDL-C, mg/dL	51 (43–62)	51 (43–62)	51 (44–61)	0.764
LDL-C, mg/dL	73 (54–98)	80 (55–102)	71 (51–98)	0.489
non-HDL-C, mg/dL	102 (77–126)	104 (80–126)	98 (77–126)	0.381
TG, mg/dL	121 (90–169)	121 (93–169)	120 (90–166)	0.976

Note: Data are expressed as median (interquartile range). Abbreviations: LVDD, left ventricular diastolic dysfunction; FGF23, fibroblast growth factor 23; Cr, creatinine; eGFR, estimated glomerular filtration rate; HbA1c, hemoglobin A1c; TC, total cholesterol; HDL-C, high-density lipoprotein cholesterol; LDL-C, low-density lipoprotein cholesterol; TG, triglycerides.

**Table 3 biomedicines-11-01523-t003:** Transthoracic echocardiographic parameters.

Variable	Total N = 71	Without LVDD N = 46	With LVDD N = 25	*p* Value
LVEF, %	56 (54–62)	56 (53–62)	56 (54–60)	0.827
LVESV, mL	31 (28–35)	31 (28–35)	33 (29–35)	0.353
LVEDV, mL	45 (43–48)	45 (43–48)	47 (45–49)	0.155
LVMI, g/m^2^	98 (83–110)	92 (81–107)	109 (100–121)	0.003
LA volume, mL	50 (43–63)	49 (42–61)	56 (48–77)	0.166
LAVI, mL/m^2^	27 (22–33)	27 (21–31)	30 (25–37)	0.158
TAPSE, mm	22 (20–25)	22 (21–25)	21 (19–24)	0.247
RVOT proximal diameter, mm	32 (30–34)	32 (31–34)	32 (30–35)	0.565
IVSs, mm	16 (15–17)	16 (15–16)	17 (16–19)	0.014
IVSd, mm	11 (10–12)	11 (10–12)	13 (11–13)	0.002
average E/e’ ratio	9 (7–11)	8 (7–10)	11 (8–15)	0.002

Note: Data are expressed as median (interquartile range). Abbreviations: LVDD, left ventricular diastolic dysfunction; LVEF, left ventricular ejection fraction; LVESV, left ventricular end systolic volume; LVEDV, left ventricular end diastolic volume; LVMI, left ventricular mass index; LA, left atrium; LAVI, left atrial volume index; TAPSE, tricuspid annular plane systolic excursion; RVOT, right ventricular outflow tract; IVSs, interventricular septum thickness at end-systole; IVSd, interventricular septum thickness at end-diastole.

**Table 4 biomedicines-11-01523-t004:** Two-dimensional speckle-tracking echocardiography data.

Variable		Total N = 71	Without LVDD N = 46	With LVDD N = 25	*p* Value
GLS, %	apical 4 chamber	−18.6 (−21.6, −16.5)	−19.5 (−21.6, −17.2)	−17.7 (−20.9, −15.6)	0.189
	apical 2 chamber	−19.4 (−22.5, −15.6)	−20.6 (−23.3, −16.9)	−18.4 (−19.6, −15.6)	0.060
	apical 3 chamber	−19.6 (−21.7, −14.1)	−19.9 (−21.5, −15.6)	−18.6 (−21.8, −13.2)	0.373
	average	−19.1 (−21.6, −16.1)	−19.9 (−22.0, −16.3)	−18.2 (−19.6, −16.1)	0.115
GCS, %	epicardial	−9.2 (−11.7, −9.2)	−8.8 (−11.2, −6.8)	−10.2 (−12.1, −7.6)	0.348
	mid-wall	−14.9 (−17.2, −11.1)	−15.1 (−17.1, −11.1)	−14.9 (−17.2, −11.8)	0.824
	endocardial	−24.8 (−29.0, −18.5)	−24.3 (−29.7, −18.8)	−25.3 (−27.6, −18.4)	1.000
	average	−16.4 (−18.8, −11.8)	−16.2 (−18.9, −11.8)	−16.4 (−18.4, −13.5)	0.962

Note: Data are expressed as median (interquartile range). Abbreviations: LVDD, left ventricular diastolic dysfunction; GLS, global longitudinal strain; GCS, global circumferential strain.

## Data Availability

The data presented in this study are available in the main article.

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
