# Peer review of "Fibroblast Growth Factor 23: Potential Marker of Invisible Heart Damage in Diabetic Population"

_biomedicines, 2023, doi:10.3390/biomedicines11061523_

Round 1

Reviewer 1 Report

the manuscript titled "Fibroblast Growth Factor 23: Potential Marker of Invisible Heart Damage in Diabetic Population" describes a biostatistics study in which authors evaluated different parameters able to predict the cardiovascular risk.

It is a very well-written manuscript highlighting the use of computational/statistical techniques useful in scientific research. I think that a concise figure highlighting all the steps of research must be inserted to favour readiness.

The study would also benefit of a PCA analysis well-constructed in a radar-like graph.

Some minor points related to quality of figures: please revise and add high resolution figures.

Author Response

Thank you very much for such an informative review. It was a pleasure to read it. We revised our manuscript and did our best to fulfill your expectations.

We modified a study flow chart (figure 5) by adding principal diagnostic methods in order to improve readiness.

As for the PCA analysis, it is a sophisticated technique, however, it is unfortunately unavailable for us in such a short time. Instead, we prepared a 3D surface chart (please find an attached file) showing the relationship between FGF23, average GLS and epicardial GCS.

We checked all the figures, and they seem to be of good quality. Other reviewers did not have any concerns. Please let us know exactly which figures need an improvement.

Thank you very much for your time and significant remarks.

Kind regards,

Anna Kurpas with my colleagues

Reviewer 2 Report

Reviewing the manuscript entitled, “Fibroblast Growth Factor 23: Potential Marker of Invisible Heart Damage” by Kurpas A et al., this is an article focusing on relevance between FGS23 and diastolic dysfunction of the heart using GCS in DM patients. Although this is an interesting research, Questions remain about the evaluation method as a clinical study. The authors need to respond to the following concerns.

In this study, as a patient background, you basically target long-term elderly DM patients who have no obvious ischemic heart disease and who are in a stable condition with treatment. In this case, it is thought that the non-LVDD subjects do not have early symptoms of heart failure. Is that correct? On what grounds was the presence or absence of LVDD, two groups? Reading the manuscript, it looks like they were divided into two groups as a pilot study.

The manuscript states that the LVDD diagnosis was performed according to clinical practice guidelines. However, the description of specific numerical values for each group such as E/e' is important. The authors should modify the manuscript.

Upon reading the introduction section, subject appears to be DM patients with heart failure. Why is the item not including BNP or NT-proBNP?

Both the LVDD and non-LVDD groups showed an EF of 56%, a result that can be judged as a slight decrease in contractility. The authors should state why contractile dysfunction is not considered. Is this an early symptom of IHD in DM patients?

Despite being diagnosed as non-LVDD, the epicardial GCS of patients with long-term DM was significantly lower than normal, and FGS23 was significantly elevated. How does the authors interpret this result? The author should discuss this point deeply in the discussion section. Should early cardiac events in DM be considered HEpEF-like findings rather than IHD?

DM tends to cause peripheral arterial disease, and as described in the introduction section, it is a well-known fact that diabetes is associated with IHD. The evidence for the absence of ischemic heart in the subject is sparse. The authors should include items for EF and left ventricular wall motion abnormalities in the patient background.

The presence or absence of arterial hypertension is also insufficient. It is necessary to describe the specific average value of each group.

Molecular biology description of FGS23 is sparse. In particular, the authors should address medical speculation regarding the link between fibrosis and FGS23.

The authors should add the abbreviation table for reading easily. 

Minor editing of English language required. There is no particular problem.

Author Response

Thank you very much for such an informative review. We revised our manuscript and did our best to fulfill your expectations.

Our study included only clinically stable patients with no symptoms of heart failure or ischemic heart disease. Diagnosis of LVDD was based on the transthoracic echocardiography findings.

We added data concerning average E/e’ ratio. Please find them in the table 3.

All subjects accounted for outpatients who were clinically stable and enjoyed wellbeing. They did not present specific symptoms of heart failure and ischemic heart disease. Therefore, we did not analyze BNP/NT-proBNP value.

We excluded patients after the myocardial infarction from a study group, in order to avoid overlapping of left ventricular injury caused by ischemic heart disease and heart failure. Patients enrolled in our study had no regional wall motion abnormalities in transthoracic echocardiography and no symptoms of ischemic heart disease. Otherwise, a multilayer analysis of myocardial deformation might have appeared unreliable.

The following text was added to a discussion section: “Data concerning FGF23 level in a diabetic population are rather poor and inconsistent, especially in terms of patients with preserved renal function [54,55]. In the literature, there are a few hypotheses explaining the relationship between elevated FGF23 and diabetes mellitus [56-63].”.  

Furthermore, we added a new paragraph at the end of Discussion section which concludes discussion.

We excluded patients with ischemic heart disease. As we mention above, patients enrolled in our study had no regional wall motion abnormalities in transthoracic echocardiography and no symptoms of ischemic heart disease. Neither coronary angiography nor coronary computed tomographic angiography was performed. Our findings highlight the importance of early detection of heart failure with preserved ejection fraction. It is said that prognosis of patients with reduced and preserved left ventricular ejection fraction is similar.

According to a diary of self-control, individuals with no previous history of arterial hypertension had blood pressure lower than 140/90 mmHg, whereas patients with a previously diagnosed arterial hypertension, who took antihypertensive drugs, had blood pressure lower than 130/80mmHg. Patients enrolled in our study were well controlled in terms of diabetes mellitus and cardiovascular risk factors.

We add a following sentence to Materials and Methods section: “Diaries of self-control were used to verify home blood pressure monitoring.”

We added an information about the relationship between FGF23 and fibrosis in the Introduction section.

Please find an attached file with an abbreviation table.

The paper has been carefully revised by a native English speaker who is a retired Professor of History in the United States.

Thank you very much for your time and significant remarks.

Kind regards,

Anna Kurpas with my colleagues

Reviewer 3 Report

This study aimed to determine whether fibroblast growth factor 23 (FGF23) may be used as a blomarker of early myocardial damage among patients with long-standing diabetes mellitus type 2 (T2DM) and no previous history of myocardial infarction. Total 71 patients with a median age of 70 years were enrolled and underwent conventional endocardiography and two-dimensional speckle tracking echocardiography and serumFGF23 measurement. The results indicated that a cardiac strain is a reliable tool for subtle myocardial damage assessment. The myocardial injury in T2DM patients occured to be layer-specfic and started from the epicardium. An elevated level of FGF23 was significantly associated with a reduced value of epicardial GCS (r=0.2438; p=0.041), but both parameters showed no significant difference between patients with LVDD and without LVDD. There are some concerns as listed in the followin:

(1) Typos and others:

P3/19

(a) 49% of the patients were hospitalized due to CVD : 15%?

(b) Table 1 67,5; 29,7: 67.5; 29.7

P5/19 (a) LBEDV: LVEDV

P8/19

Fig. 3 r=0,2438; p=0,041: r=0.2438; p=0.041

(2) Inconsistent writing format for References (chect all)

(a) Authors: e.g. R36, R37, R42

(b) Page numer: R16 (no number); R24 (470-483), R26, R30, R31, R36-39, R44 vs. R53 (4957-60)

Author Response

Thank you very much for your review. We revised our manuscript and did our best to fulfill your expectations.

While preparing the revised version of manuscript, we considered all your comments.

Thank you very much for your time and significant remarks.

Kind regards,

Anna Kurpas with my colleagues

Round 2

Reviewer 2 Report

This version reaches to an acceptable quality. Congrats.

Author Response

Thank you very much for your reply. As you implied, I did not make any changes to the manuscript. It was pleasure to read your review.

Kind regards, 

Anna Kurpas